# Dual-View Single-Shot Multibox Detector at Urban Intersections: Settings and Performance Evaluation

**DOI:** 10.3390/s23063195

**Published:** 2023-03-16

**Authors:** Marta Lenatti, Sara Narteni, Alessia Paglialonga, Vittorio Rampa, Maurizio Mongelli

**Affiliations:** 1CNR-IEIIT, 10129 Turin, Italy; 2Department of Control and Computer Engineering (DAUIN), Politecnico di Torino, 10129 Turin, Italy

**Keywords:** smart mobility, object detection, video content analysis, single-shot multibox detector

## Abstract

The explosion of artificial intelligence methods has paved the way for more sophisticated smart mobility solutions. In this work, we present a multi-camera video content analysis (VCA) system that exploits a single-shot multibox detector (SSD) network to detect vehicles, riders, and pedestrians and triggers alerts to drivers of public transportation vehicles approaching the surveilled area. The evaluation of the VCA system will address both detection and alert generation performance by combining visual and quantitative approaches. Starting from a SSD model trained for a single camera, we added a second one, under a different field of view (FOV) to improve the accuracy and reliability of the system. Due to real-time constraints, the complexity of the VCA system must be limited, thus calling for a simple multi-view fusion method. According to the experimental test-bed, the use of two cameras achieves a better balance between precision (68%) and recall (84%) with respect to the use of a single camera (i.e., 62% precision and 86% recall). In addition, a system evaluation in temporal terms is provided, showing that missed alerts (false negatives) and wrong alerts (false positives) are typically transitory events. Therefore, adding spatial and temporal redundancy increases the overall reliability of the VCA system.

## 1. Introduction

Nowadays, the *smart city* paradigm is changing the asset of the urban environment thanks to the rapid growth of digital technologies and communication infrastructures. By interconnecting people and things, smart cities scenarios provide more efficient, fast, ubiquitous, and accessible services to citizens [1]. In this context, *smart mobility* applications are empowered by the high speed and low latency properties of 5G networks [2], being suitable for ensuring road safety [3] and monitoring dangerous situations [4]. The huge amount of sensor data and the availability of fast computing resources at the edge of the 5G networks have paved the way to advanced deep learning (DL) models for real-time video content analysis (VCA) scenarios [5].

Both real-time localization and object classification methods from video streams are mandatory requirements for VCA solutions. To this aim, different DL architectures based on convolutional neural networks (CNNs) have recently been proposed [6]. However, among the most widely exploited approaches, you-only-look-once (YOLO) and single-shot multibox detectors (SSD) algorithms stand out for their performance and computing efficiency [7]: the former is indeed one of the fastest and most accurate networks for real-time object detection [8], while the latter is a benchmark for real-time multi-class object detection at different scales [9].

In this paper, we consider a driver alert scenario, where an urban intersection is monitored by two cameras and an SSD-based object detection model is trained to identify, localize and, eventually, signal the presence of obstacles to public transportation vehicles approaching the surveilled area. Particular focus will be given to investigating the advantages of using two cameras instead of a single one, in terms of object detection and alert generation performance. To this purpose, the VCA model will be evaluated using qualitative and tailored quantitative approaches, exploiting both spatial and temporal redundancy.

The paper is organized as follows. First, we discuss relevant literature on the topic. Then, we recall the SSD-based method adopted, and we thoroughly describe the on-field implementation. Finally, we present and discuss the results in terms of object detection performance and the related alert generation performance.

## 2. Related Works

Object detection and/or tracking via multiple camera sensors is a widespread topic in computer vision research. Multi-view 3D object recognition [10] consists in reducing complex 3D object classification tasks to simpler 2D classification tasks by rendering 3D objects into 2D images. Real objects are surrounded by cameras posed at different viewpoints with configurations leading to multi-view proposals, such as MVCNN [11], GVCNN [12], View-GCN [13] and RotationNet [14] architectures. These methods use the most successful image classification networks, i.e., VGG, GoogleNet, AlexNet, ResNet, as backbone networks. Then, global 3D shape descriptors are obtained by aggregating selected multi-view features through approaches that account for both content and spatial relationships between the views.

Transfer learning approaches prove extremely useful, especially when dealing with scarcely available data. To this end, several open source datasets for object detection in urban traffic optimization and management have recently become available. These datasets focus either on pedestrian or vehicle tracking and detection, combining inputs from multiple cameras and extending visual coverage (e.g., [15,16]).

An overview of recent multi-camera solutions for object detection is presented below. In [17], a novel multi-view region proposal network that infers the vehicles position on the ground plane by leveraging multi-view cross-camera scenarios is presented, whereas an end-to-end DL method for multi-camera people detection is studied in [18]. In [19], a vehicle detection method that applies transfer learning on two cameras with different focal length is proposed. The processing consists of two steps: first, a mapping relationship between input images from the cameras is calculated offline through a robust evolutionary algorithm; then, CNN-based object detection is performed online. More specifically, after a vehicle region is detected from one camera, it is transformed into a binary map. This map is then used to filter CNN feature maps computed for the other camera’s image. It is important to outline that finding the relationship between the two cameras is crucial to solve the problem of duplicated detection, as different cameras may focus on the same vehicles. The same problem is raised in [20], where the authors present a novel edge-AI solution for vehicles counting in a parking area monitored via multiple cameras. They combine a CNN-based technique for object localization with a geometric approach aimed at analyzing the shared area between the cameras and merging data collected from them. Multi-camera object detection is also investigated in [21], which presents an autonomous drone detection and tracking system exploiting a static wide-angle camera and a lower-angle camera mounted on a rotating turret. In order to save computational resources and time, the frame coming from the second camera is overlaid on the static camera’s frame. Then, a lightweight version of YOLOv3 detector is developed to perform the object detection. Another recent work on multi-camera fusion for CNN-based object classification [22] devised three fusion strategies: early, late and score fusion. A separate CNN was first trained on each camera. Afterward, feature maps were stacked together and processed either from the initial layers (early fusion) or at the penultimate layers (late fusion). In addition, score fusion was performed, by aggregating the softmax classification scores in three possible ways: by summing, or by multiplying, the scores across cameras, or by taking the maximum score across them. Results showed that late and score fusion led to an accuracy improvement, with respect to early fusion and single camera proposals. Multi-camera detection has gained increasing importance in several areas besides smart mobility applications. For example, several solutions have recently been proposed in the area of fall detection for remote monitoring of fragile patients. In [23], multi-camera fusion is performed by combining models trained on single cameras together into a global ensemble model at the decision-making level, providing higher accuracy with respect to local single-camera models and avoiding computationally expensive cameras calibration. The dual-stream fused neural network method, proposed in [24], first trains two deep neural networks to detect falls by using two single cameras and then merges the results through a weighted fusion of prediction scores. The obtained results overcome the existing methods in this domain.

All these proposals deal with high-intensity computational methods, while, on the contrary, real-time field-deployable applications impose computational complexity constraints as well. To solve this key issue, we propose here a simple but effective dual-view fusion and detection method and compare its performance with real field experiments [25]. In particular, our solution exploits a transfer learning approach, which consists in training the object detection model on a single camera, in updating it through an additional training by feeding the other camera’s images, and then by fusing the single detection signals to generate alerts at the decision level. This speeds up the overall training time and saves computational resources, with respect to other existing decision-making level camera fusion approaches, such as [22,23].

## 3. Video Content Analysis System

### 3.1. Single-Shot Multibox Detector Model

The SSD network is composed of a *backbone* stage for feature extraction and a *head* stage for determining the output. The backbone is a feature pyramid network (FPN) [26], which is a CNN able to extract feature maps representing objects at different scales. It comprises a bottom-up pathway connected to a top-down pathway via lateral connections. The SSD head is a sequence of output maps, which determines the output of the network in the form of bounding box coordinates and object classes. Additionally, the SSD network exploits the concept of *priors* (also known as *anchor boxes*), a special kind of box whose predefined shape can guide the network to correctly detect objects of the desired class.

The SSD head is composed of multiple output maps (grids) with different sizes. Each grid decomposes the image into cells, and each cell expresses whether or not it belongs to a particular object, in terms of bounding box coordinates and object class. Lower resolution output maps (i.e., smaller size grids), having larger cells, can detect larger scale objects; in contrast, larger size output grids, having denser cells, are used to predict smaller objects. The use of multiple outputs improves the accuracy of the model significantly, while maintaining the ability to predict objects in real time.

#### 3.1.1. Loss Function

The training of the SSD model is based on the minimization of the following loss function L:(1)$$L=L_{loc}+L_{conf}+L_{boxiness},$$
where $L_{loc}$ evaluates the object localization of the model, $L_{conf}$ evaluates the object classification ability and $L_{boxiness}$ term refers to the *boxiness*, i.e., the ability of discriminating boxes from background throughout SSD output grids.

Considering object localization, we define $y_{gt}=\left(x,y,w,h\right)$ as the ground truth box coordinates vector for a generic object, with x,y expressing box center coordinates, *w* the box width and *h* the box height. Similarly, we denote with $y_{pr}=\left(x_{pr},y_{pr},w_{pr},h_{pr}\right)$ the predicted box coordinates vector for that same object. A discrepancy between the real and predicted box positions is measured by the vector $a≐|y_{gt}-y_{pr}|$, with coordinates $\left(a_{1},a_{2},a_{3},a_{4}\right)=(|x-x_{pr}|,|y-y_{pr}|,|w-w_{pr}|,|h-h_{pr}\left|\right)$.

The $L_{loc}$ term is then computed through the pseudo-Huber loss function [27]:(2)$$L_{loc}=\sum_{i=1}^{4}\delta^{2}\;\left(\sqrt{1+{\left(\frac{a_{i}}{\delta }\right)}^{2}}-1\right),$$
with δ being a fixed quantity that controls the steepness of the function. The pseudo-Huber loss provides the best performance, with minimal computational costs with respect to the Huber and other types of loss functions [28]. In this study, δ was set to 1.5, following preliminary training runs.

Referring to object classification, let $y_{c}$ be the true class label for each class c=1,…,N, where *N* is the number of classes. Additionally, let ${\hat{p}}_{c}$ be the corresponding class probability estimates. The second loss term, $L_{conf}$, is then a cross-entropy loss, computed as follows:(3)$$L_{conf}=-\sum_{c=1}^{N}y_{c}\;log\left({\hat{p}}_{c}\right)$$

After prediction, the SSD model also outputs an estimate of the boxiness, expressed as a real value $b_{pr}\in \left[0,1\right]$, which can be interpreted as the model confidence in recognizing whether any object is present in each cell of the network output grids. Consequently, the quantity $b_{bg}=1-b_{pr}$ defines the level of confidence of each cell to be part of the background.

The last term $L_{boxiness}$ relies on a focal loss function [29], which is chosen for its ability to penalize the false positives, i.e., the background points wrongly detected as objects by the model. The boxiness loss $L_{boxiness}$ is then computed as
(4)$$L_{boxiness}=-\left[\alpha \;b_{bg}^{\gamma }log\left(b_{pr}\right)+\left(1-\alpha \right)\;b_{pr}^{\gamma }\;log\left(b_{bg}\right)\right],$$
where the parameter α acts as a weight for those cells being covered by a box and 1−α acts as weight for the background cells; the parameter γ controls the shape of the function. Higher values of γ require lower loss values to better distinguish boxes from background (i.e., to have $b_{pr}>0.5$). The attention of the model is thus devoted to the harder-to-detect samples.

#### 3.1.2. Network Parameters, Training and Testing

Non-maximum suppression (NMS) [30] was performed to refine the predictions of the model. Indeed, it may often occur that multiple boxes are predicted for the same ground truth object. The NMS algorithm filters out the predicted boxes based on the class confidence and the intersection over union (IoU) method [31] between them. In particular, for a given SSD output grid and class, for each real object, the predicted box (if any) with the highest class confidence is picked. This box is then chosen as a reference to compute the IoU between itself and all the other predicted boxes, keeping only those with a value below a threshold. In our case, we fixed this threshold at 0.1. Choosing such a low value allows to filter out boxes characterized by even small overlaps with the reference one, therefore reducing the presence of false positives.

Table 1 summarizes the properties and parameters of the SSD model adopted in this work. The choice of the SSD output grids dimensions was guided by a preliminary analysis on a range of suitable values, performed to individuate a proper balance between model accuracy and computational complexity. Additionally, the selection of regions of interest from the foreground area, as better detailed in the scenario definition, required lower sized grids, able to capture bigger foreground objects. The network is trained to recognize three classes of objects: ‘vehicle’, ‘rider’, and ‘pedestrian’.

### 3.2. VCA Architecture

We define here the main pipeline of the VCA system for alert generation, whose inference and training/retraining flowcharts are sketched in Figure 1. The first pipeline (Figure 1a) sketches the object detection blocks employed to generate alerts (inference phase) by exploiting image fusion on both cameras. The second pipeline (Figure 1b) focuses on retraining the baseline SSD by adding *TLC2* images via transfer learning, thus obtaining a final model, i.e., SSD_ret_. More specifically, the inference block diagram shows the real-time processing pipeline adopted to generate the alarm signal $A_{L}$ by fusing together the single-view alerts $A_{L1}$ and $A_{L2}$ produced by the alert generation blocks $AG_{1}$ and $AG_{2}$ that are fed by the output of the SSD_ret_ object detectors attached to the single camera TLC1 and TLC2, respectively. The two cameras have a broad field of view, but in order to define the area of potential danger to be monitored, a region of interest (ROI) is determined and adapted for each camera. The alert $A_{L}$ is then employed to alert the driver by activating visual and acoustic alarms on the bus console. The final inference stage $A_{L}$ is designed to integrate the two independent outputs of the single alert generators related to each camera view and to perform information fusion at the decision level with the aim of increasing the overall reliability and accuracy of the system.

Figure 1b shows the retraining procedure adopted to update the baseline SSD network of the single-view system (that uses only TLC1 data) by including also images from the TLC2 camera. In fact, the baseline SSD model (i.e., the green block in Figure 1b) is preliminarily trained on a set of images extracted from three open-source datasets (Open Images Dataset [32], ETH Pedestrian Dataset [33] and EuroCity Dataset [34]) that contain annotated images of urban traffic scenes. Afterward, the images captured by *TLC1* were added to these datasets to complete the training of the baseline SSD model. To further improve the flexibility, reliability and, in particular, the detection accuracy of the VCA system, the baseline SSD model was later retrained on a set of 10,000 additional images acquired from the *TLC2* camera. The term retraining refers to the procedure of updating the parameters of a previously trained model based on the addition of new data by transfer learning methods [35]. From now on, we will refer to the final retrained model as SSD_ret_ (blue block in Figure 1b). The generalizing capabilities of the baseline SSD and the retrained SSD_ret_ models were assessed using a test dataset consisting of frames extracted from a 1-h video, for both cameras. Both videos were first synchronized and cut to align the start and end time stamps, then converted from the h263 format to the mp4 format using the FFmpeg tool [36] (with compression factor 1.25). Finally, 1000 frames were extracted for each recording.

### 3.3. Data Labeling

YOLOv5x [37], one of the state-of-the-art YOLO networks for object detection in real-time applications, was adopted to define ground truth boxes, i.e., to label the objects actually present in each image. For this purpose, YOLOv5x was applied on each image of the training, retraining, and test datasets in order to recognize objects of the classes ‘Car’, ‘Bus’, ‘Truck’, ‘Motorcycle’, ‘Bicycle’, and ‘Pedestrian’. Then, these classes were grouped into three more generic classes, namely ‘Vehicle’, ‘Rider’, and ‘Pedestrian’. Ground truth boxes were provided in the YOLO format (x_center_, y_center_, width, height), and subsequently converted in the SSD format (x_min_, y_min_, width, height). The results of this automatic labeling step were then manually inspected to verify the presence of sufficiently accurate ground truth boxes. In the presence of detection errors inside the monitored area, the corresponding images were removed from the dataset. Based on the ground truth boxes, we also defined the number of ground truth alerts, which were raised any time at least one ground truth box was detected within the ROI.

## 4. Driver Alert Use Case

### 4.1. Scenario Definition

Piazza Caricamento is one of the locations with the highest concentration of pedestrian and road traffic in the historic center of Genoa, Italy, as it connects the east and west areas of the city and, above all, it is located nearby the main tourist attractions (e.g., the aquarium, the pedestrian area on the harbor, and the most important architectural and artistic sites of the city). The area monitored by the proposed VCA system is the intersection between the pedestrian area of the harbor, the vehicular access to the parking lot, and the access roads to the underground tunnel below Piazza Caricamento corresponding to the latitude and longitude coordinates 44.4110720922656 N, 8.928069542327654 E (expressed in decimal degrees). A dedicated public transportation bus lane, which is characterized by limited visibility, interconnects with the monitored intersection. The area is often crowded with pedestrians and vehicles frequently passing through to access the car parking. Hence, potential collisions with buses coming from their dedicated lane represent a real risk scenario that makes Piazza Caricamento a suitable location to implement a VCA system. The proposed solution consists in an automatic system able to detect the presence of pedestrians and/or vehicles inside the area via VCA processing, and to generate an appropriate alert to the bus approaching the intersection. Real-time monitoring is performed via two Bosh DINION IP Bullet 6000I HD, 2, 8–12 MM cameras, which are professional surveillance HD cameras compliant with the SMPTE 296M-2001 standard [38] and ONVIF profiles G and S [39] to guarantee the interoperability with the AI components. We will refer to these cameras as *TLC1* and *TLC2*.

As previously noted, only objects within each ROI of the cameras can generate an alert to be sent to the driver. As a result, the two ROIs strictly overlap. Since our SSD model involves multiple output grids, the ROI was resized for each of them based on their dimension. Figure 2 displays the fields of view covered by the two cameras and reports the selected ROIs for each adopted grid.

As further emphasized in the following sections, the main goal of our work is to understand to what extent the joint use of two cameras can represent an added value for the VCA task with respect to the use of a single camera (either *TLC1* or *TLC2*).

### 4.2. Performance Evaluation

Two types of performance figures will be considered to evaluate the VCA monitoring system, namely *object detection performance*, that is the ability of the system to correctly identify different classes of objects inside the ROI, and *alert generation performance*, that is the ability of the system to trigger an alert if and only if at least one object is present in the monitored area.

For the sake of simplicity, the system performance results were assessed considering only 3 grids (i.e., 12 × 20, 6 × 10 and 3 × 5) with priors of size 1 × 2 (more suitable for identifying people) and priors of size 2 × 1 (more suitable for identifying vehicles).

Finally, the VCA system performance results were evaluated also in terms of computation time required for object detection and alert generation. The average inference time per frame was assessed locally on a host equipped with an Intel Core i5 dual-core processor at 2.6 GHz, 8 GB RAM memory banks, and running the macOS 10.15.7 operating system.

#### 4.2.1. Object Detection Performance

The ability of each component of SSD_ret_ (according to the aforementioned grids and priors) to identify objects of different classes inside the ROI was evaluated by calculating the average confusion matrix over the whole test dataset, for each camera, namely the average number of correctly identified objects (TP_obj_), the average number of undetected objects (FN_obj_), and the average number of objects detected but not actually present in the ground truth image (FP_obj_). The obtained values were then compared with the average number of real objects per image. Then, in order to measure the object detection performance from a comprehensive point of view, precision (PRE_obj_) and recall (REC_obj_) were assessed for each considered frame, both individually for single grids and priors, and aggregating all outputs. Precision measures the number of correctly identified objects to the total number of detected objects, whereas recall measures the number of correctly detected objects to the total number of ground truth objects. These metrics were then averaged across all the frames in the test dataset (i.e., 1000 frames).

#### 4.2.2. Alert Generation Performance

The ability of SSD_ret_ to generate alerts when an object is inside the ROI was assessed by calculating the confusion matrix over the entire test dataset, considering two possible outputs of the system, namely the presence of an alert (*alert* = 1) or its absence (*alert* = 0), for each input image. The following elements of the confusion matrix were considered: the total number of correctly generated alerts (TP_alert_), the total number of ground truth alerts not triggered by the system (FN_alert_), the total number of alerts incorrectly triggered by the system (FP_alert_), and the total number of non-alert situations in which the alert is correctly not triggered by the system (TN_alert_). It is also important to underline that, in light of the technological implementation of the alerts triggering system of each camera, incorrect alerts (either FN_alert_ or FP_alert_) were only triggered when no true positives had already been generated for the same image.

As previously described, SSD models provide different outputs from output maps of different sizes. Therefore, system performance was first evaluated by considering alerts detected individually by each grid and prior and then by evaluating the total amount of alerts identified by the aggregation of all grids and all priors. Alert generation performance was evaluated both individually on the two cameras (*TLC1* and *TLC2*, separately) and then on their fusion. In the latter case, an alert is generated when at least one of the two cameras detects an object within the ROI.

Since the frames considered in our use case are temporally continuous, we also decided to evaluate if the presence of FN_alert_ and FP_alert_ could be considered a transient phenomenon or not. Hence, we computed also the FN${}^{*}$_alert_ and FP${}^{*}$_alert_, representing the false negatives and false positives occurred at least in two consecutive frames. Any FN_alert_ or FP_alert_ events present in just one frame were therefore considered spurious and avoided by waiting for the next frame before performing inference.

## 5. Results

### 5.1. Object Detection Performance

A base model was trained on a set of images composed by *TLC1* images and external images from open-source datasets on mobility scenarios. The base model was then retrained on a dataset extracted from *TLC2* recordings yielding SSD_ret_. The procedure of retraining (on *TLC2* images only) an already pre-trained model offers several advantages over training from scratch (using *TLC1* and *TLC2* images). Notably, retraining was faster than the full training. Specifically, the time required to retrain the model was more than 10 times shorter than the original training time of the baseline SSD (i.e., 42 h). Table 2 reports the obtained object detection performance for each camera, each grid, and each prior separately in terms of mean confusion matrix over the entire test dataset. Average precision and average recall were also computed.

According to Table 2, it appears that the *TLC1* images contain fewer ground truth objects inside the ROI than the TLC2 ones. However, no ground truth events filmed by *TLC2* are captured by the 3 × 5 grid with 1 × 2 prior. Hence, it was not possible to calculate TP_obj_, FN_obj_ and recall in that case. Since the number of false positives is on average higher than the number of false negatives, PRE_obj_ is lower than REC_obj_, except when considering a 12 × 20 grid with 1 × 2 prior. In addition, we can observe how grids with a larger number of cells (i.e., 12 × 20 and 6 × 10) are generally able to detect more objects than the smallest grid (i.e., 3 × 5). This may be due to the fact that objects within the ROI are typically in the background and thus more easily detected by denser grids, characterized by smaller cell sizes.

The global object detection performance results of SSD_ret_ on both cameras were then evaluated in terms of precision and recall, reported in Table 3. These values were obtained by considering all the grids and priors used to define the model’s architecture (as defined in Table 1). *TLC1* yielded a low precision of about 17% and a satisfying recall, equal to about 90%. In contrast, *TLC2* yielded a much higher precision of about 73% and recall similar to *TLC1* (i.e., about 89%).

### 5.2. Alert Generation Performances

Alert generation performance was first evaluated separately on the two cameras and then considering the fusion between the alerts generated by the two, as shown in Table 4. The results reported in Table 4 are consistent with those shown in Table 2 since grids with a larger number of cells (i.e., 12 × 20 and 6 × 10) are able to generate more alerts than the smallest grid (i.e., 3 × 5). In particular, with the exception of the 3 × 5 grid that mostly detects vehicles, most of the alerts seem to be raised by objects that correspond to the prior of size 1 × 2 (i.e., pedestrians in the ROI). From these results, we can observe that both the number of ground truth alerts and the number of correctly predicted alerts (TP_alert_) increase when considering the data fusion of both cameras (*fusion(TLC1,TLC2)*), compared to the individual *TLC1* and *TLC2*. Figure 3 shows an example of an alert correctly detected by *TLC2* but not by *TLC1*. This image would therefore constitute an FN event considering only *TLC1*, but it is correctly classified as a TP event when *fusion(TLC1,TLC2)* is considered.

If we focus, for example, on the 12 × 20 grid and the 1 × 2 prior (Table 4), we can observe that *TLC1* alone detects 62 ground truth alerts (54 TP_alert_), while *TLC2* detects 41 ground truth alerts (27 TP_alert_) and *fusion(TLC1,TLC2)* detects 76 ground truth alerts (61 TP_alert_). These results confirm how different grids and priors are able to identify different objects, and consequently generate different alerts. For this reason, we finally evaluated the global alert generation performance results, obtained by combining all the outputs provided by different priors and grids and by considering the temporal continuity of the frames. The results of this global evaluation are reported in Table 5.

The estimated average elapsed time during the inference phase for the whole alert generation process on a single camera is about 0.46 s per frame, while the elapsed time of the decision fusion is about $1.8\cdot 10^{-6}$ s and may be neglected. Thus, the total inference time of the multi-camera VCA system (not parallelized) is about 0.92 s per frame.

## 6. Discussion

A VCA monitoring system based on a SSD architecture was implemented and evaluated in terms of its ability to detect objects in the surveilled area and its related ability to generate alerts. Specifically, the VCA system foresees possible dangerous situations inside a intersection through the use of a multi-camera deep learning-based object detection system. The choice to merge data at the decision level was motivated by its simplicity, which allows to operate within the time constraints dictated by a real-time application. In addition, the system built in this way can easily compensate for the lack of one of the two possible inputs, ensuring robustness against possible failures or damages to the system.

Comparing the *TLC1* and *TLC2* cases, it can be seen that the former has a rather low precision in detecting objects. This result is further confirmed by the performance results of alert generation (Table 5). Provably, the precision of *TLC1* in terms of alert generation is lower than the corresponding *TLC2* precision (i.e., 62% and 77%, respectively). As a result, *fusion(TLC1,TLC2)* reaches a higher precision (i.e., 69%) with respect to *TLC1* alone. In contrast, the recall of *TLC1* in terms of alert generation is slightly higher than the corresponding *TLC2* recall (i.e., 86% and 80%, respectively). As a result, *fusion(TLC1,TLC2)* yields a higher recall (i.e., 84%) with respect to *TLC2* alone. In summary, by combining the two cameras, there is a significant increase in precision with respect to *TLC1* alone and a slight improvement in recall compared to *TLC2*. The monitoring system based on SSD_ret_ yields quite satisfactory alert generation accuracy when considering a single camera (i.e., about 94%).This means that the retraining phase did not erase what the model learned from *TLC1* images, i.e., there is no catastrophic forgetting [40]. Although accuracy remains almost stable (93%) when considering *fusion(TLC1,TLC2)*, the introduction of a second camera *TLC2* improves the overall safety by allowing the identification of a higher number of real dangerous situations (i.e., 125 ground truth alerts) within the area of interest. In fact, the combination of *TLC1* and *TLC2* enables the triggering of 40% more ground truth alerts than *TLC1* alone. The increase in the number of alerts is mainly due to the different framing of the two cameras, and thus the increased field of view of the object detection system. Consequently, also the absolute number of TP_alert_ increases (from 77 to 105) after the outputs of the two cameras are merged. Since we are dealing with a highly unbalanced dataset, where the number of dangerous situations is considerably lower than the number of safe situations, it could be useful to evaluate the F1-score. Specifically, it can be seen that the use of two cameras results in an F1-score of 75%, which is higher than that obtained by using *TLC1* alone (i.e., 73%).

By using not only spatial redundancy, i.e., the different views of the same monitored area captured by *TLC1* and *TLC2*, but also the temporal continuity of the frames, we can design a post-processing algorithm that uses the information of two or more consecutive video frames instead of a single one as assumed so far. In this case, the actual output alert signal is generated if it is triggered by at least two consecutive frames. By exploiting the temporal continuity, the amount of wrong predictions is reduced, as indicated in Table 5, where FP${}^{*}$_alert_ and FN${}^{*}$_alert_ (i.e., the number of FPs and FNs persisting in at least two consecutive frames) are consistently lower than FP_alert_ and FN_alert_, respectively. This reduction in the number of false and missed alarms proves that FPs and FNs are generally spurious events that can be easily removed by considering a certain time window. However, it is worth noting that this method introduces a one-frame delay in the alert signal generation stage.

In addition, a local evaluation of the total inference time per frame was performed, demonstrating the ability of the proposed multi-camera VCA system to generate the alert in a sufficiently short time (less than 1 s), which is compatible with the system requirements to make a decision in real time. However, more precise evaluations will be needed following specific on-site deployment.

This study presents some limitations. First of all, the multi-camera system was evaluated using a single fusion technique directly applied at the decision level. In future studies, different data fusion techniques, including early and late fusion at different depths of the network, should be compared to evaluate possible further improvements in terms of the system reliability. Moreover, although the network was originally trained on a heterogeneous set of images from the experimental test-bed (*TLC1*) and open source datasets, the dataset used for retraining SSD_ret_ included only *TLC2* frames captured in daytime. Therefore, it will be necessary to evaluate the system’s ability to generalize in different scenarios, such as its robustness in different weather and light conditions (e.g., day/night and sunny/rainy weather). Lastly, at the current stage, possible security issues following malicious attacks on the main components of the system (e.g., cameras, onboard units, and edge servers) have not been considered yet. In particular, the alert generation system could be vulnerable to adversarial attacks aimed at changing the output of the system, which could cause potential dangerous situations. In the future, it will be necessary to devise robust solutions to these types of attacks, such as considering the introduction of a Bayesian layer in the vision system [41].

## 7. Conclusions

This work focuses on the development and evaluation of an single-shot multibox detector-based object detection system applied to an urban scenario. In particular, we evaluated the effectiveness of adding a second camera (*TLC2*) in terms of detecting potential hazardous situations within the region of interest. The introduction of a second camera, in addition to the first one (*TLC2*) not only makes the video content analysis system more robust with respect to possible failures due to *TLC1* malfunctions but also leads to a higher number of correctly detected alarms thanks to a wider coverage of the surveilled area. Furthermore, the number of false negative (FN-type) events is reduced by considering temporal continuity in successive frames. In the specific smart mobility use case, FN-type errors were considered to be more important than false positive (FP-type) errors. Indeed, the number of negative events misclassified as positive (i.e., FP-type), will result in alarms that do not correspond to the presence of objects or obstacles in the region of interest. Such errors are considered less critical because they simply cause unnecessary alerts to be sent, without endangering the driver. However, in the long run, these redundant alarms may make the driver less confident in the system’s ability to correctly identify dangerous situations. Future studies will focus on further validation of the proposed solution. Finally, the formalization of an algorithm that can leverage the temporal continuity provided by videos, instead of relying on individual frames, could be investigated.

## Figures and Tables

**Figure 1 sensors-23-03195-f001:**
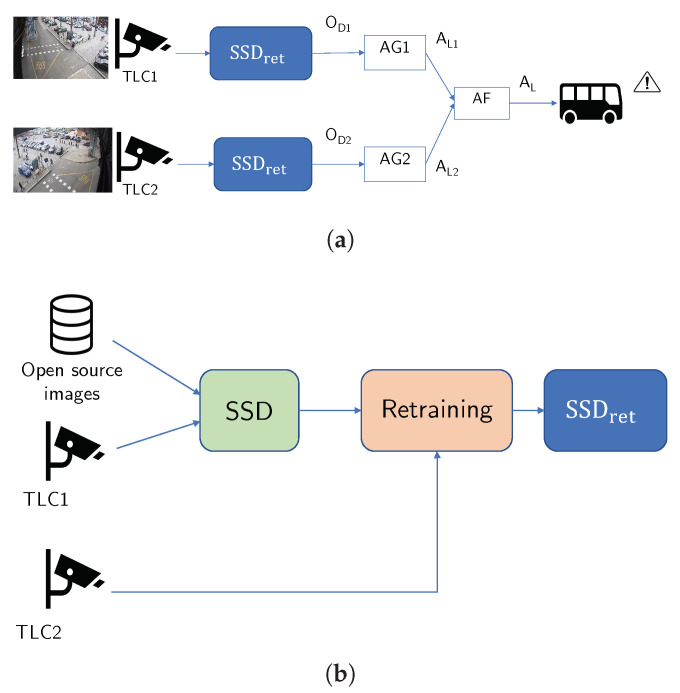
Flowchart of the procedures exploited for the proposed VCA system, sketching the alert generation and fusion (**a**) based on model SSD_ret_, obtained via a retraining process (**b**). (**a**) Inference. (**b**) Retraining.

**Figure 2 sensors-23-03195-f002:**
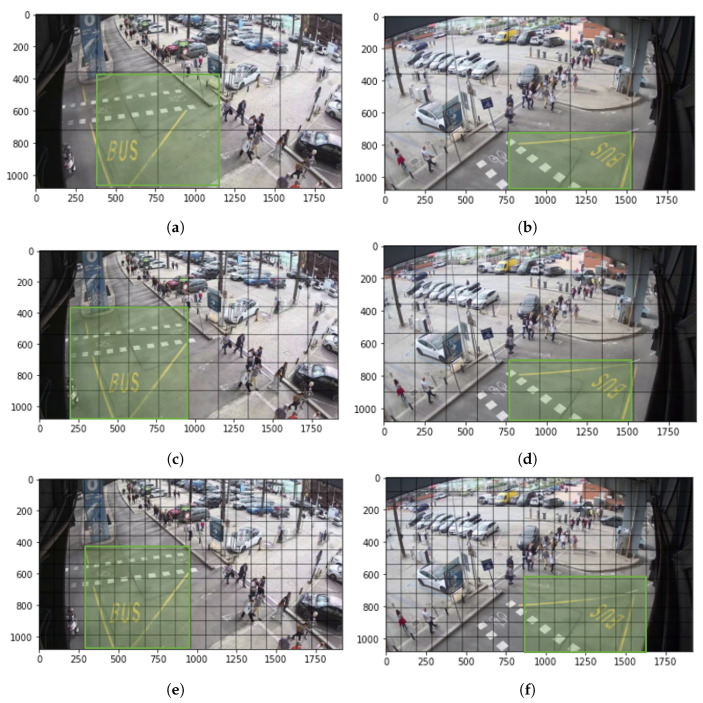
Regions of interest (ROIs) inside the monitored area (green rectangles), for each considered SSD output grid on *TLC1* (left column) and *TLC2* (right column). (**a**) ROI on *TLC1* for output grid of size 3 × 5. (**b**) ROI on *TLC2* for output grid of size 3 × 5. (**c**) ROI on *TLC1* for output grid of size 6 × 10. (**d**) ROI on *TLC2* for output grid of size 6 × 10. (**e**) ROI on *TLC1* for output grid of size 12 × 20. (**f**) ROI on *TLC2* for output grid of size 12 × 20.

**Figure 3 sensors-23-03195-f003:**
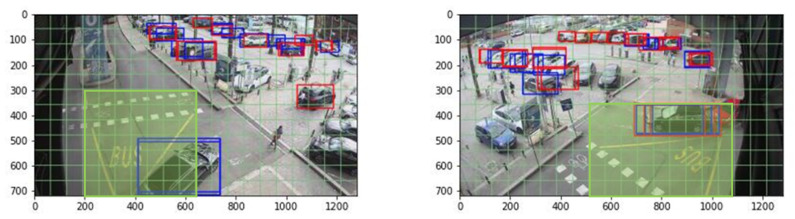
Example of the same object correctly detected within the ROI (green area) by *TLC2* (**right**), but missed by *TLC1* (**left**). Ground truth boxes are shown in blue, while predicted boxes are shown in red.

**Table 1 sensors-23-03195-t001:** Parameters and properties of the adopted SSD.

Output grids	24 × 40, 12 × 20, 6 × 10 and 3 × 5
Priors	1 × 1, 2 × 1, 4 × 1, 1 × 4 and 1 × 2
# trainable parameters	5000
Learning rate	$10^{-4}$
δ	1.5
α	0.85
γ	2
IoU threshold	0.1

**Table 2 sensors-23-03195-t002:** Mean and standard deviation (between parentheses) of TP_obj_, FP_obj_, FN_obj_ and percentage of PRE_obj_ and REC_obj_ for each camera, grid, and prior of the SSD_ret_ model.

	TLC1						TLC2					
	**#Real** **Objects**	**TP_obj_**	**FP_obj_**	**FN_obj_**	**PRE_obj_**	**REC_obj_**	**#Real** **Objects**	**TP_obj_**	**FP_obj_**	**FN_obj_**	**PRE_obj_**	**REC_obj_**
Grid: 12 × 20 Prior: 1 × 2	0.19 (0.80)	0.10 (0.55)	0.04 (0.20)	0.08 (0.87)	55%	54%	0.71 (1.43)	0.24 (0.79)	0.22 (0.57)	0.40 (0.96)	43%	31%
Grid: 12 × 20 Prior: 2 × 1	0.02 (0.31)	0.02 (0.24)	0.08 (0.36)	0.005 (0.13)	11%	66%	0.34 (1.18)	0.24 (0.99)	0.28 (0.65)	0.10 (0.61)	24%	67%
Grid: 6 × 10 Prior: 1 × 2	0.05 (0.35)	0.05 (0.30)	0.15 (0.42)	0.02 (0.23)	19.76%	63.46%	0.13 (0.48)	0.07 (0.36)	0.07 (0.27)	0.06 (0.28)	37%	43%
Grid: 6 × 10 Prior: 2 × 1	0.005 (0.08)	0.005 (0.10)	0.18 (0.47)	0.00 (0.00)	1.6%	100%	0.08 (0.40)	0.01 (0.14)	0.08 (0.35)	0.07 (0.43)	15%	21%
Grid: 3 × 5 Prior: 1 × 2	0.01 (0.14)	0.00 (0.05)	0.07 (0.25)	0.01 (0.13)	4.11%	42.86%	-	-	1.70 (0.59)	-	0%	-
Grid: 3 × 5 Prior: 2 × 1	0.001 (0.03)	0.00 (0.00)	0.08 (0.28)	0.001 (0.03)	0%	0%	0.07 0.33	0.04 0.23	1.63 0.56	0.03 (0.19)	1.3%	61.44%

**Table 3 sensors-23-03195-t003:** Global object detection performance of SSD_ret_ for each camera by considering all the grids and priors as defined in Table 1. Precision: PRE_obj_; Recall: REC_obj_.

	TLC1	TLC2
PRE_obj_	17%	73%
REC_obj_	90%	89%

**Table 4 sensors-23-03195-t004:** Number of ground truth alerts and TP_alert_ for each grid and prior using single camera processing (*TLC1*, *TLC2*) and data fusion of both cameras (*fusion(TLC1,TLC2)*).

	TLC1		TLC2		Fusion(TLC1,TLC2)	
	**Ground Truth Alerts**	**TP_alert_**	**Ground Truth Alerts**	**TP_alert_**	**Ground Truth Alerts**	**TP_alert_**
Grid: 12 × 20 Prior: 1 × 2	62	54	41	27	76	61
Grid: 12 × 20 Prior: 2 × 1	9	6	29	25	34	28
Grid: 6 × 10 Prior: 1 × 2	66	49	29	23	76	57
Grid: 6 × 10 Prior: 2 × 1	8	8	3	0	11	8
Grid: 3 × 5 Prior: 1 × 2	3	2	2	0	5	2
Grid: 3 × 5 Prior: 2 × 1	7	4	3	0	10	4

**Table 5 sensors-23-03195-t005:** Ground truth alerts, TP_alert_, TN_alert_, FP_alert_, FN_alert_, FP${}^{*}$_alert_ and FN${}^{*}$_alert_ obtained from all grids and priors, on single cameras (*TLC1*, *TLC2*) and their fusion (fusion(TLC1,TLC2)).

	Ground Truth Alerts	TP_alert_	TN_alert_	FP_alert_	FN_alert_	FP${}^{*}$_alert_	FN${}^{*}$_alert_
TLC1	89	77	865	46	12	2	0
TLC2	74	59	908	18	15	1	2
fusion(TLC1,TLC2)	125	105	827	48	20	3	3

## Data Availability

The data presented in this study may be available upon request to the corresponding author.

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
