# Peer review of "Dual-View Single-Shot Multibox Detector at Urban Intersections: Settings and Performance Evaluation"

_sensors, 2023, doi:10.3390/s23063195_

Round 1

Reviewer 1 Report

Thank you for the manuscript on the multi camera video content analysis system using SSD network. There are few recommendations to further improve the content for readability.

·      Please use traffic engineering terminology instead of cross roads and study location should be appropriately explained.

·      Literature from lines 58-78 can be enhanced. Also please incorporate more recent studies in the related works.

·      Research gap in last paragraph needs to be clarified. Also, instead of referring to section as that in reports, please directly mention the flow of work or how it was achieved. Please avoid report style. A revision is request in this regard.

·      Figure 1 caption needs to be concise.

·      Please revise lines 172-173 and 207-208, 217-219 in view of above comment.

·      Is the information in the foot note aligned with the journal guidelines?

·      Section 4.2 states "the main objective of this work… To this end, two types of performance will be considered.” This is needs to be revised.

·      Please further elaborate the limitation of this study.

Round 2

Reviewer 1 Report

Overall authors have satisfactorily addressed previous comments. Following are a few minor adjustments. Please avoid referring to sections (a common element in thesis/technical reports). See the following.

·      “As previously described in Section 3.1, SSD models provide different outputs from 310 output maps of different sizes.”

·      “This section first reports the results of the object detection for single cameras.”

·      Lines 42 to 45

Author Response

We would like to thank the Reviewer whose suggestions helped us to improve again the overall quality of the manuscript. 

As suggested, we have removed references to sections and report like sentences (lines 42-45, 117-119, 179, 232-234, 309, 323-325, 327,389).